# Effectiveness of short-term isothermic-heat acclimation (4 days) on physical performance in moderately trained males

**Jake Shaw**[1☯]*, **Cory Walkington**[1☯], **Edward Cole**[1☯], **Damien O. Gleadall-Siddall**[1,2‡], **Rachel Burke**[1,3‡], **James Bray**[1‡], **Andrew J. Simpson**[1‡], **Rebecca V. Vince**[1], **Andrew T. Garrett**[1]

1 Department of Sport, Health, and Exercise Science, University of Hull, Hull, United Kingdom, 2 School of Life Sciences, Coventry University, Coventry, United Kingdom, 3 Research Scientist, Johnson & Johnson, London, United Kingdom

☯ These authors contributed equally to this work.
‡ DOGS, RB, JB and AJS also contributed equally to this work.
* j.shaw-2013@hull.ac.uk

## Abstract

### Introduction

A typical heat acclimation (HA) protocol takes 5–7 d of 60–90 minutes of heat exposure. Identifying the minimum dose of HA required to elicit a heat adapted phenotype could reduce financial constraints on participants and aid in the tapering phase for competition in hot countries. Therefore, the aim of this study was to investigate a 4 d HA regimen on physical performance

### Methods

Twelve moderately trained males were heat acclimated using controlled hyperthermia ($T_{re}$>38.5°C), with no fluid intake for 90 min on 4 consecutive days, with a heat stress test (HST) being completed one week prior to (HST2), and within one-week post (HST3) HA. Eleven completed the control study of HST1 versus HST2, one week apart with no intervention. Heat stress tests comprised of cycling for 90 min @ 40% Peak Power Output (PPO); 35°C; 60%RH followed by 10 minutes of passive recovery before an incremental test to exhaustion. Physical performance outcomes time to exhaustion (TTE), PPO, end rectal temperature ($T_{re\ END}$), and heart rate ($HR_{END}$) was measured during the incremental test to exhaustion.

### Results

Physiological markers indicated no significant changes in the heat; however descriptive statistics indicated mean resting $T_{re}$ lowered 0.24°C (-0.54 to 0.07°C; $d = 2.35$: very large) and end-exercise lowered by 0.32°C (-0.81 to 0.16; $d = 2.39$: very large). There were significant improvements across multiple timepoints following HA in perceptual measures; Rate of perceived exertion (RPE), Thermal Sensation (TS), and Thermal Comfort (TC) (P<0.05). Mean

**Data Availability Statement:** Data can be found on Figshare (DOI:10.6084/m9.figshare.19762828).

**Funding:** The author(s) recieved no specific funding for this work. All equipment and research was conducted at the University of Hull.

**Competing interests:** The authors have declared that no competing interests exist.

TTE in the HST increased by 142 s (323±333 to 465±235s; $P$ = 0.04) and mean PPO by 76W (137±128 to 213±77 W; $P$ = 0.03).

## Conclusion

Short-term isothermic HA (4 d) was effective in enhancing performance capacity in hot and humid conditions. Regardless of the level of physiological adaptations, behavioural adaptations were sufficient to elicit improved performance and thermotolerance in hot conditions. Additional exposures may be requisite to ensure physiological adaptation.

## Introduction

Exercise in excessive heat can cause greater physiological and perceptual stress compared to mild conditions [1–3] resulting in diminished exercise capacity [4]. Heat acclimation (HA) regimes are used in conjunction with exercise to stress thermoregulatory and cardiovascular systems [5], in order to achieve a heat-adapted phenotype more capable of tolerating and exercising in hot conditions [6, 7].

Heat acclimation is achieved by increasing core temperature ($T_c$) over a sustained period. A sufficient stimulus is fundamental in generating a physiological strain elicit adaptation [6, 8, 9], with the magnitude of adaptation seemingly dependant on the scale and regularity of thermal strain and impulse applied [2]. In this case, the HA method used to achieve this is the controlled hyperthermia method. Controlled hyperthermia is executed based on endogenous measures, namely $T_c$ [6, 10, 11], and can provide continuous training by achieving specific and individualised $T_c$ through a combination of passive and active HA [6, 12]. As a participant begins to adapt to the heat, their adaptive threshold becomes more difficult to attain. Therefore, the balance between work and rest needs to be altered to ensure consistency, or a progression of endogenous heat strain to continue the adaptive response [6]. The balance between work and rest to specify and reach a certain $T_c$ guarantees a consistency, or a progression of endogenous heat stress to achieve adaptation–although this requires modifications in application during each session. By comparison, fixed intensity methods are very simple to implement, with participants sustaining a fixed workload throughout each individual active acclimation session [13–25]. It can become problematic to attain and sustain a target $T_c$ of 38.5°C using the more conventional continuous HA methods, nonetheless a controlled isothermic HA methodology defeats this problem by making sure that the target $T_c$ is achieved and then sustained using passive and active heat stress [2].

Many studies have explored hydration status of participants exercising in the heat and it is generally accepted that hydration status influences physiological and performance responses in the heat [26]. Naturally, during exercise in the heat dehydration occurs. As a result, fluid-regulatory, cardiovascular, and thermal strain are increased [10]. Restricting fluid intake exacerbates the strain applied on the body, encouraging cardiovascular and fluid regulatory adaptations that are key mechanisms of a heat-adapted phenotype [26–31]. One of these key mechanisms is the expansion of plasma volume facilitated through sodium retention and elevated intra-vascular albumin levels [10, 32] encourages the body to retain and utilise the bodily fluid it already has without external replenishment.

In recent years, researchers have focused on the minimum dose required to produce these physiological and behavioural adaptation with the aim of improving performance in the shortest timeframe [2, 33]. Utilising the controlled hyperthermia methodology protocols vary in

exercise mode, exposure duration, exposure intensity, and the number of repeated bouts. Heat acclimation adaptations are achieved within 4–7 d [34] and a 5 d protocol is commonly used [2, 5, 10, 28, 33, 35–37]. Limited evidence exists following 4 d of heat exposure. Further, due to heterogeneity in study designs, comparisons of HA after 4 d and the typical 5 d protocol are challenging [38–40].

Identifying the minimum required HA dose is beneficial for health and logistical reasons. A shorter HA protocol may reduce the risk of injury or heat illness and the reduced exposure time can be implemented more easily during the tapering phase of an athletes training shortly before competition [2]. This may provide financial benefits from one less day of training in an environmental chamber. The aim of this study was therefore to explore the effectiveness of 4 consecutive 90 min, isothermic, HA bouts with no fluid intake by assessing physiological and subjective markers of adaptation, as well as performance markers pre- and post-intervention. The hypothesis was that 4-consecutive days of short-term isothermic heat acclimation (STHA) would enhance physiological markers of adaptation, whilst perceived heat stress would be reduced and, subsequently, performance would improve.

## Materials and methods

### Experimental design

Twelve moderately trained males who trained 2–3 times weekly took part in this study, completing 4 consecutive days of short-term isothermic HA with no fluid intake during each bout (Fig 1). Twelve completed the intervention trial, conducted one-week before and within one week of STHA intervention. Eleven completed the control trial (HST1 v HST2) conducted one week apart with no intervention. Participants completed a ramped protocol to exhaustion to calculate 40% of the PPO to be used as the resistance for the HSTs.

### Participants

Participants were fully informed of all experimental procedures by both written and verbal means. Experimental periods were conducted outside of the British summer time. Heat stress tests and acclimation bouts were conducted at the same time of day in the mornings. Participants were asked to refrain from strenuous exercise 24 h prior to each HST as well as caffeine and alcohol 12 h before each bout of exercise. Participants were instructed to wear a shirt and shorts and wear similar if not the same clothing on each visit. Pre-exercise medical questionnaires and informed consent forms were completed before being recruited and all were in good health. Ethical approval was provided by the University of Hull's Ethics Committee (No. 1516177) following World Health Organization Declaration of Helsinki guidelines. With HST resistance determined via PPO, participants were categorised accordingly in line with the recommendations of De Pauw and colleagues [41] (n = 5: PL1, PPO <280 W; n = 2: PL2, PPO 280–319 W; n = 5: PL3, PPO 320–379 W). Furthermore, participants consisted of n = 4 runners, n = 5 cyclists, n = 1 weightlifter, and n = 2 footballers who trained 2–3 times a week.

### Aerobic fitness test

Participants performed an incremental ramp protocol on a cycle ergometer (Daum Electronic Gmbh, Furth, Germany) to determine peak oxygen uptake ($\dot{V}O_{2\,peak}$) and PPO (W) starting at 50 W. Resistance increased by 25 W every minute until volitional exhaustion. Breath by breath expired air was collected via metabolic cart system (Cortex Metalyzer 3B, Cortex Biophysic, Leipzig, Germany) calibrated by 3 L calibration syringe (Hans Rudolph 3 L, Cranlea & Co., Birmingham, UK) and calibration gas (5% $CO_2$, 15% $O_2$, Cranlea & Co., Birmingham, UK).

**Fig 1. Schematic model of the short-term isothermic heat acclimation protocol for moderately trained males.**

Participants' rate of perceived exertion (RPE) and HR were recorded every minute. Peak oxygen uptake was determined via a rolling 30 s average, therefore the final $\dot{V}O_{2\ peak}$ value was the final 30 s before exhaustion. Peak power output was determined by the power the participant achieved before exhaustion. All participants received verbal encouragement in the waning stages of the test. Termination occurred when either the participants voluntarily ended the test, or the participant could not maintain >60 rpm.

**Short-term isothermic heat acclimation protocol.** The experiment took place in an Environmental Chamber (Type SSR 60-20H, Design Environmental, Gwent, Wales) in 40°C and 60%RH for 90 minutes per day using controlled hyperthermia with no fluid intake. Environmental conditions were recorded every 10 min. Participants cycled (Monark 824E, Monark Exercise AB, Varberg, Sweden) until reaching target $T_{re}$ of 38.5°C as quickly as possible and resistance was individually adapted as necessary every 5 min to maintain thermal stimulus. Upon achieving this target, participants ceased exercising and were seated for the remainder of the session–unless $T_{re}$ fell below 38.5°C then light exercise was prescribed. This protocol is identical to previous work conducted by Garrett and colleagues [5, 10, 35, 36] but over 4 d HA.

**Hydration status.** Participants were instructed to provide a pre-exercise nude body mass ($BM_{nude}$) measure prior to samples being given. Post-exercise $BM_{nude}$ was obtained after capillary blood samples were taken and before a urine sample was given. Both $BM_{nude}$ measures were obtained on every visit and on days one and four of acclimation. Urine samples were collected on days one and four of acclimation and on every visit for HST's following the same

routine as HA measures. Urine specific gravity ($SG_u$) was calculated using a refractometer (Unicron-N, Urine specific gravity refractometer, Atago CO., Tokyo, Japan) [42]. Urine colour ($colour_u$) was measured using a $colour_u$ chart [43] and urine osmolality ($osm_u$) was measured using an osmometer (Model 3320, Advanced Instruments Inc., Massachusetts, USA). All measures were collected immediately prior to and post exercise, in duplicate and analysed immediately with the mean value reported.

**Blood.** Venous and capillary blood samples were obtained pre- and post-heat exposure on day one and day four of the HA. Additional capillary samples were obtained pre and post each HST. Venous samples were taken from an antecubital vein and capillary blood samples from a fingertip. Venous blood was collected in one 4 mL heparin "green top", one 4 mL serum separating tube "gold top", and one 4 mL Ethylenediaminetetraacetic acid "purple top" (VACU-ETTE©, Greiner Bio One Ltd, Stonehouse, United Kingdom). Collection tubes were immediately inverted a minimum of eight times to ensure a homogenous sample. Approximately 1 mL of whole blood from heparin tubes was removed and analysed using an automated blood gas analyser (ABL800 Basic analyser, Radiometer, UK) within 30 minutes of collection for $[Na^+]$, $[Cl^-]$ and $[K^+]$. Serum separating tubes were allowed to clot at room temperature. Collection tubes were then centrifuged (Heraeus Labofuge 400R, Kendro Laboratory Products, Bishops Stortford, UK) at 2118g for 10 minutes at 4˚c. After centrifugation, ~1 mL of plasma/serum from each tube was aliquoted into cryo-tubes and stored at -80˚c until analysis. Samples were allowed to thaw at room temperature and vortexed before analysis. An automated benchtop analyser (ABX Pentra 400, Horiba UK, Northampton, UK) was used to quantify [Total Protein] and [Albumin] from serum samples. Enzyme-linked immunosorbent assays were used to quantify aldosterone (Aldosterone ELISA Kit [ab136933], abcam, Cambridge, UK); cortisol (Cortisol ELISA Kit [ab108665], abcam, Cambridge, UK) and HSP70 (HSP70 High Sensitivity ELISA Kit [ab133061]). The manufacturer intra-assay coefficient of variation for all participant samples in duplicate was 4.4–6.6% for aldosterone, ≤9% for cortisol, and 5.9–11.4% for HSP70. The recommended protocols provided by the manufacturer were followed for each ELISA. Changes in percentage of plasma volume (%PV) was calculated from changes in haemoglobin (Hb) and haematocrit (Hct), as defined by Dill and Costill [44]. Both Hb and Hct were analysed in duplicate from capillary blood samples with the mean reported. Samples were analysed using a Hb analyser (Hemocue 201+, Radiometer Ltd, Crawley, UK) and a microhematocrit centrifuge (Hawksley & Sons, Lancing, UK) respectively.

## Heat stress test

The HSTs were conducted in the same environmental chamber set to 35˚C, 60%RH and consisted of 90 min continuous exercise on a cycle ergometer (Daum Electronic Gmbh, Furth, Germany) using individualized workloads at 40% PPO output achieved in the $\dot{V}O_{2\ peak}$ trial. Upon completion, there was a 10 min passive recovery period before performing an incremental ramp protocol to exhaustion. Increments were 2% of PPO applied every 30 s, commencing from the individualised 40% PPO workload. Recorded measures from the performance trial included End $T_{re}$ (˚C), time to exhaustion (TTE) (s), power (W), and End HR (b·min$^{-1}$). Environmental conditions were recorded using the same procedure as the acclimation sessions. Capillary blood, urine samples and $BM_{nude}$ were obtained, pre and post exercise on every visit.

**Heart rate.** Heart rate was measured using a HR monitor (Polar FS1, Polar Electro, OY, Finland) at baseline and every 10 min throughout the continuous exercise trial. End exercise HR was taken at exhaustion of the incremental protocol to exhaustion.

**Body temperature.** Core body temperature was measured using a rectal thermistor (U Thermistor, Grant Instruments Ltd, Cambridge, UK) inserted 10cm past the anal sphincter.

Skin thermistors (Type EUS-U-V5-V2, Grant Instruments Ltd, Cambridge, UK) were placed on the body in four sites as outlined by Ramanathan [45]. From these, mean skin temperature ($\bar{T}_{sk}$) and mean body temperature ($\bar{T}_b$) were calculated [45]. Temperature data was recorded at baseline and every 10 min using a portable data logger (2020 series data logger, Grant Instruments Ltd, Cambridge, UK).

## Perceptual measures

Participants were asked to give their RPE (6–20) [46], thermal comfort (TC) (1–5) and thermal sensation (TS) (1–13) [47] These were collected every 10 min from rest to completion of the 90 min steady state exercise bout.

## Data analysis

The stress response of dependant measures in STHA and HSTs were analysed through SPSS (IBM SPSS Statistics, Version 25, IBM Corp, Armonk, New York, USA). The Shapiro-Wilk test determined normal distribution. A two-way repeated measures ANOVA was used to determine main effects between day one and day four of HA, pre vs post HSTs as well as inter-action and effect over time for all dependant measures. If a participant did not complete the 90 min steady state exercise component during the HST, their final measurement was duplicated into missing time points to include all participants. Pairwise comparison, least significant difference (LSD) correction *t*-tests were used when appropriate to determine significance at specific time-points. Significance was defined at <0.05 where appropriate to define variation among groups through SPSS. The change in thermal markers on day one to day four of acclimation were analysed using one-way ANOVA, with repeated measures and LSD correction, pairwise comparison *t*-tests to isolate differences between days. Where appropriate, data is reported as mean±SD with 95% confidence intervals (95%CI) and the magnitude of effect using Cohen's *d* effects sizes (0.2–0.59 small; 0.6–1.19 moderate; 1.2–1.99 large, 2.0–4.0 very large) [48]. Intraclass Correlation Coefficient estimates and their 95% confident intervals were calculated using SPSS based on a mean-rating ($k = 3$), absolute agreement, 2-way mixed effects model (Table 1).

## Results

Twelve ($n = 12$) male participants (mean±SD; age: 35±15yrs; height: 175.3±4.5 cm; mass: 79.7 ±11.2 kg; $\dot{V}O_{2\,peak}$: 47.2±9.9 mL·kg$^{-1}$·min$^{-1}$) who took part in training at least twice a week completed this study. All twelve participants completed a pre- and post-intervention HST and the four consecutive days of acclimation. Whereas $n = 11$ completed the control study. Data is representative of all twelve participants unless otherwise specified.

### Control study

The HST1 versus HST2 (n = 11) was a control trial completed 1 week apart with no intervention (Fig 1). Table 1 contains Intraclass Correlation Coefficient calculations for HST 1–2 for all resting and end-exercise measures.

### Intervention study

The HST2 versus HST3 was an intervention study completed one week prior (HST2) and within one week (HST3) of 4 consecutive days of 90 min (40˚C; 60%RH) using controlled hyperthermia with dehydration stimuli (Fig 1).

**Table 1. Intraclass Correlation Coefficient calculations for resting and end-exercise measures in SPSS using single-rating, absolute-agreement, 2-way random effects model ($n$ = 11).**

| Variable | | Intraclass Correlation | 95% Confidence Interval | | F Test with True Value 0 | | | |
|---|---|---|---|---|---|---|---|---|
| | | | Lower Bound | Upper Bound | Value | df1 | df2 | Sig. |
| HR R | Single Measures | 0.773 | 0.382 | 0.932 | 8.193 | 10 | 10 | 0.001 |
| HR E | Single Measures | 0.660 | 0.175 | 0.893 | 5.504 | 10 | 10 | 0.006 |
| MST R | Single Measures | 0.347 | -0.189 | 0.756 | 2.237 | 10 | 10 | 0.110 |
| MST E | Single Measures | -0.122 | -0.452 | 0.394 | 0.700 | 10 | 10 | 0.709 |
| MBT R | Single Measures | 0.455 | -0.200 | 0.820 | 2.545 | 10 | 10 | 0.078 |
| MBT E | Single Measures | -0.110 | -0.716 | 0.520 | 0.814 | 10 | 10 | 0.625 |
| $T_{re}$ R | Single Measures | 0.363 | -0.285 | 0.779 | 2.090 | 10 | 10 | 0.130 |
| $T_{re}$ E | Single Measures | 0.402 | -0.218 | 0.793 | 2.319 | 10 | 10 | 0.100 |
| OSMO R | Single Measures | 0.243 | -0.320 | 0.707 | 1.692 | 10 | 10 | 0.210 |
| OSMO E | Single Measures | 0.369 | -0.126 | 0.763 | 3.320 | 10 | 10 | 0.036 |
| USG R | Single Measures | 0.205 | -0.329 | 0.682 | 1.578 | 10 | 10 | 0.242 |
| USG E | Single Measures | 0.280 | -0.163 | 0.702 | 2.240 | 10 | 10 | 0.110 |
| COL R | Single Measures | 0.000 | -0.399 | -1.326 | 0.514 | 10 | 10 | 0.500 |
| COL E | Single Measures | 0.356 | -0.184 | 0.761 | 2.275 | 10 | 10 | 0.106 |
| BM R | Single Measures | 0.997 | 0.987 | 0.999 | 875.210 | 10 | 10 | 0.000 |
| BM E | Single Measures | 0.998 | 0.994 | 1.000 | 1548.805 | 10 | 10 | 0.000 |
| Hb R | Single Measures | 0.694 | 0.199 | 0.907 | 5.296 | 10 | 10 | 0.007 |
| Hb E | Single Measures | 0.479 | -0.098 | 0.824 | 2.886 | 10 | 10 | 0.055 |
| Hct R | Single Measures | 0.627 | 0.075 | 0.884 | 4.170 | 10 | 10 | 0.017 |
| Hct E | Single Measures | 0.924 | 0.754 | 0.979 | 24.477 | 10 | 10 | 0.000 |
| RPE E | Single Measures | 0.770 | 0.339 | 0.933 | 7.153 | 10 | 10 | 0.002 |
| TS R | Single Measures | -0.027 | -0.277 | 0.405 | 0.905 | 10 | 10 | 0.561 |
| TS E | Single Measures | 0.710 | 0.254 | 0.911 | 5.957 | 10 | 10 | 0.005 |
| TC R | Single Measures | -0.019 | -0.677 | 0.586 | 0.967 | 10 | 10 | 0.521 |
| TC E | Single Measures | 0.437 | -0.199 | 0.810 | 2.476 | 10 | 10 | 0.084 |

notes: R = resting; E = End-Exercise; HR = heart rate; MST = mean skin temperature; MBT = mean body temperature; Tre = rectal temperature; OSMO = urine osmolality; USG = urine specific gravity; COL = urine colour; BM = body mass; Hb = haemoglobin; Hct = haematocrit; RPE = rate of perceived exertion; TS = thermal sensation; TC = thermal comfort

**Table 2. Mean±SD thermal stress and strain on day one and four of short-term heat acclimation for ten moderately trained males ($n$ = 10).**

| | Day 1 | Day 4 | Cohens D | p-value |
|---|---|---|---|---|
| Mean $T_{re}$ (˚C) | 38.28±0.19 | 38.14±0.16 | 0.70 | 0.06 |
| Time to $T_{re}$ 38.5˚C (min) | 37.27±6.89 | 40.90±8.13 | 1.81 | 0.09 |
| Time spent above 38.5˚C | 52.73±6.89 | 49.11±8.13 | 1.81 | 0.09 |
| Work (KJ) | 39.08±8.33 | 39.56±10.52 | 0.04 | 0.92 |
| BM Change (%) | -1.3±0.1 | -1.3±0.1 | 0.01 | 0.89 |
| %PV Change | -5.78±6.55 | -4.85±11.39 | 0.09 | 0.82 |

Ta = ambient temperature; RH = relative humidity; Tre = rectal temperature; min = minutes; KJ = kilojoules; % = percentage; PV = plasma volume; BM = body mass. Data presented as mean±SD for ten male participants.

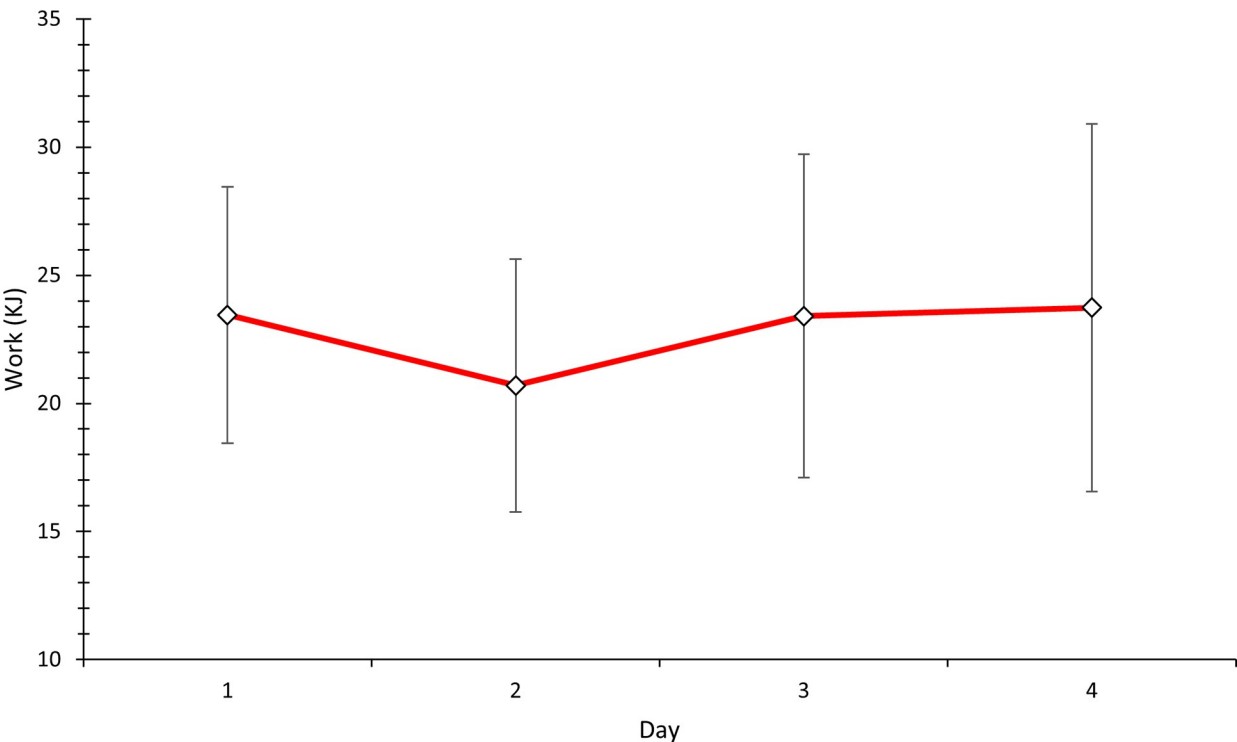

**Fig 2. Mean (SD) work output on days 1–4 of acclimation after 90 min heat exposure.**

### Acclimation

**Thermal stress and strain day 1 versus day 4.** Rectal temperature ($T_{re}$) responses are indicative of a consistent thermal strain (Table 2). There was no significant increase in mean work completed ($P = 0.92$) (Fig 2). There were no significant changes in time to reach 38.5°C ($n = 10$), however descriptive statistics indicated a 10% increase (37.27±6.89 to 40.90±8.13min; $P = 0.09$; $d = 1.81$: Large). Similarly, there was a limited concurrent decrease in time above 38.5°C ($n = 10$) of 7% (52.73±6.89 to 49.11±8.13min; $P = 0.09$; $d = 1.81$: Large) between days one and four of acclimation (Fig 3).

**Hydration status.** Urine colour, $osm_u$, $SG_u$, and BM were measured pre- and post-exercise on day one and four of acclimation (Table 3). There was a main effect across time ($F$ [1,9] = 140,798, $P<0.001$) as well as a significant interaction effect ($F$ [1,9] = 23.4, $P = 0.001$) in BM measures. Pairwise comparisons showed a significant decrease between post-measures on days one and four ($T$ [9] = 2.8, $P = 0.02$), as well as in day one pre-post ($T$ [9] = 10.9, $P<0.001$) and day four ($T$ [9] = 12.3, $P<0.001$). A significant main effect across time was detected in $colour_u$ ($F$ [1,9] = 22,959, P = 0.001) and further post-hoc analysis indicated significant decreases on day one pre-post ($T$ [9] = -3.9, $P = 0.004$) and day four ($T$ [9] = -3.3, $P = 0.009$). No significant main effects or interaction effect were indicated for $osm_u$ or No significant main or interaction effects were indicated in $SG_u$ measures.

**Blood markers.** Plasma sodium $[Na^+]_p$ analysis (Table 4) demonstrated significant main effects across time ($F$ [1,9] = 23.1, $P = 0.001$) and day ($F$ [1,9] = 9.4, $P = 0.013$). There was a significant main effect across time (n = 8; $F$ [1,7] = 22.5, $P = 0.002$) detected in $[aldo]_p$ measures (Table 4). Similarly, TP analysis (Table 4) indicated a significant effect across time ($F$ [1,9] = 68.4, $P<0.001$). Analysis of $[alb]_p$ (Table 4) measurements indicated a significant main effect across time ($F$ [1,9] = 26.7, $P = 0.001$). Heat shock protein corrected for total protein [HSP70/

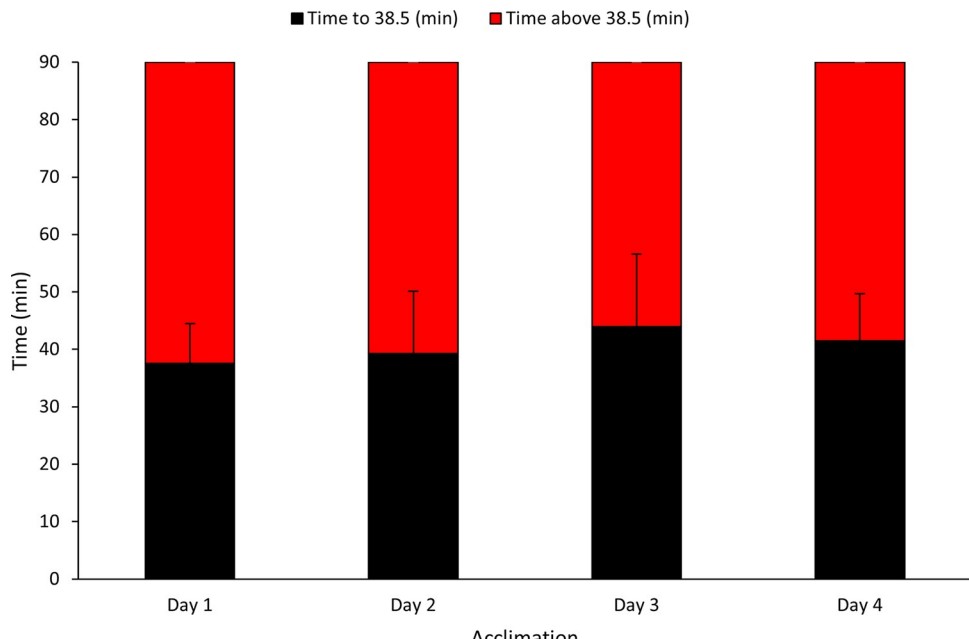

**Fig 3. Mean (SD) Time to 38.5˚C (min) and Mean (SD) Time above 38.5˚C (min) from Day 1 to Day 4 of acclimation after 90 min heat exposure.**

**Table 3. Mean±SD for BM$_{nude}$ and urinary measures of hydration (colour$_u$; osm$_u$; SG$_u$) pre- and post-exposure, on days one and four of short-term heat acclimation (*n* = 10).**

|  | Day 1 |  |  | Day 4 |  |  |
|---|---|---|---|---|---|---|
|  | **Pre** | **Post** | **p-value** | **Pre** | **Post** | **p-value** |
| BM (kg) | 76.8±6.4 | 75.0±6.3 | <0.001*& | 76.4±6.4 | 74.3±6.2 | <0.001*& |
| colour$_u$ (units) | 2±1 | 4±1 | 0.004* | 3±1 | 4±1 | 0.009* |
| osm$_u$ (mOsm/kg) | 401±333 | 465±286 | - | 424±326 | 485±263 | - |
| SG$_u$ (units) | 1.0110±0.0105 | 1.0131±0.0099 | - | 1.0109±0.0103 | 1.0125±0.0081 | - |

colouru = urine colour; osmu = urine osmolality; SGu = urine specific gravity; kg = kilograms; STHA = short-term heat acclimation

* = main effect over time; + = main effect for day; & = interaction effect. Data presented as mean±SD for ten male participants. A two-way repeated measures ANOVA and LSD correction t-tests was used when appropriate to determine differences between pre- and post-exposure, on day one and day four of STHA.

**Table 4. Mean+SD for blood measures and percentage change from pre- to post-exposure on day one and four of short-term heat acclimation.**

|  | Day 1 |  |  | Day 4 |  |  |
|---|---|---|---|---|---|---|
|  | **Pre** | **Post** | **%Change** | **Pre** | **Post** | **%Change** |
| **Adlo (n = 8)** | 441+212 | 658+306 | 49 | 565+477 | 1419+884 | 151 |
| **cortisol (*n* = 8)** | 252+102 | 279+171 | 11 | 208+120 | 282+139 | 36 |
| **HSP70 (*n* = 9)** | 6.54+0.08 | 6.52+0.06 | 0 | 6.58+0.07 | 6.52+0.07 | 0 |
| **Na+** | 140.5+2.8 | 144.6+4.7 | 3 | 139.8+3 | 141.2+4.4 | 1 |
| **TP** | 75.6+3.1 | 83.5+2.4 | 10 | 76.1+2.2 | 85.6+4.3 | 12 |
| *alb* | 757±35 | 827±45 | 9 | 763±33 | 860±84 | 13 |

[aldo]p = plasma aldosterone; pg.mL-1 = pictograms per millilitre; [Na+]p = plasma sodium; mmol.L-1 = millimoles per litre; [TP]p = total protein; mg.mL-1 = milligrams per millilitre; [cortisol]p = plasma cortisol; ug.dL-1 = micrograms per decilitre. Data is presented mean±SD for n = ten moderately trained males. A two-way repeated measures ANOVA and post-hoc LSD correction t-tests when appropriate was used to determine differences from pre- and post-exposure, on day one and day four of acclimation.

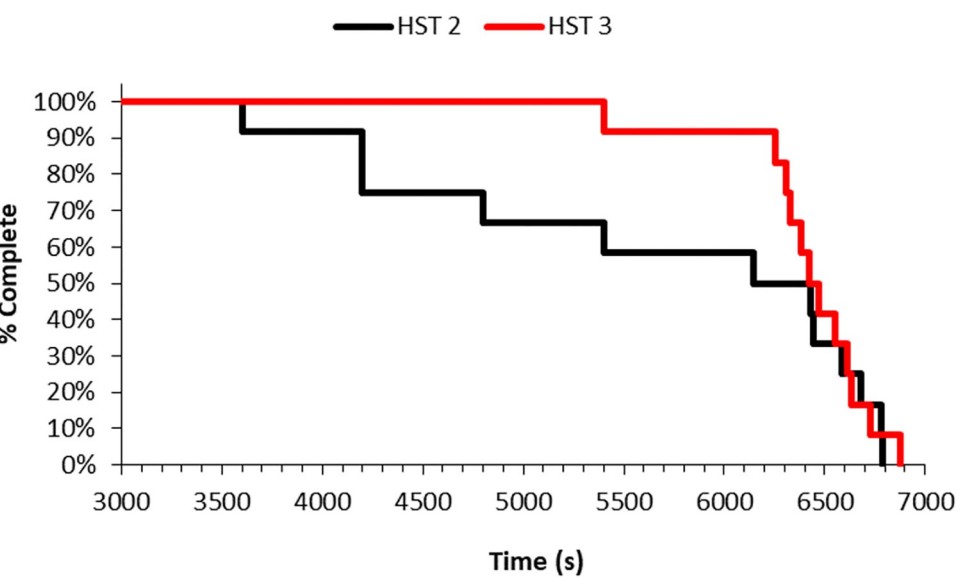

**Fig 4.** Mean±SD for $T_{re}$ (upper), $\bar{T}_b$ (middle), and $\bar{T}_{sk}$ (lower) pre- to post short-term heat acclimation in hot conditions (35˚C; 60%RH; $n$ = 12).

TP]$_p$, total protein ([TP]$_p$), and [cortisol]$_p$ analysis (Table 4) showed no significant main effect for day, time, or interaction effect from day one to four ($P$>0.05).

**Heat stress test.** *Body temperature.* There was no main effect for day in $T_{re}$ (Fig 4: upper) ($F$ [1, 11] = 3.2; $P$ = 0.103) or interaction effect ($F$ [9, 99] = 1.6; $P$ = 0.135). However, there was a significant main effect for time ($F$ [9, 99] = 165.1; $P$<0.001). Descriptive statistics indicated that mean $T_{re}$ was lower at rest by 0.24˚C (-0.54 to 0.07˚C; $d$ = 2.35: very large) and end-exercise by 0.32˚C (-0.81 to 0.16; $d$ = 2.39: very large). There was no main ($F$ [1, 11] = 2.5; $P$ = 0.145) or interaction ($F$ [9, 99] = 0.2; $P$ = 0.996) effect for day in $\bar{T}_b$ (Fig 4: middle) but there was a significant main effect for time ($F$ [9, 99] = 81.9; $P$<0.001). There were no main (F [1, 11] = 2.2; $P$ = 0.166) or interaction ($F$ [9, 99] = 1.3; $P$ = 0.246) effects for $\bar{T}_{sk}$ (Fig 4: lower).

*Heart rate and percentage change in plasma volume.* There was no main effect for day (Fig 5) ($F$ [1, 7] = 2.6; $P$ = 0.154) or interaction effect ($F$ [9, 63] = 0.7; $P$ = 0.743) in HR. However, there was a significant main effect across time ($F$ [9, 63] = 96.1; <0.001). Descriptive statistics indicated mean resting HR was lower at rest by 6 (-19 to 6 b·min$^{-1}$; $d$ = 0.32: small) and end-exercise by 10 (-23 to 4 b·min$^{-1}$; $d$ = 1.85: small). There was limited change in %PV (-4.70 to -6.63; $d$ = 0.84: moderate) pre- to post-intervention.

*Perceptual.* A main effect for day ($F$ [1, 11] = 17.3; $P$ = 0.002) and a main effect over time ($F$ [9, 99] = 105.6; $P$<0.001) was indicated for pre-post intervention RPE measures. LSD corrected post-hoc comparisons showed significant decreases at 10 ($T$ [11] = 2.3; $P$ = 0.04), 20 ($T$ [11] = 3.4; $P$ = <0.01); 30 ($T$ [11] = 3.5; $P$ = <0.01), 40 ($T$ [11] = 2.9; $P$ = 0.01), 50 ($T$ [11] = 3.9; $P$<0.01), 60 ($T$ [11] = 3.4; $P$ = <0.01), 70 ($T$ [10] = 2.8; $P$ = 0.02), and 80 ($T$ [8] = 2.5; $P$ = 0.04) minute time points from pre- to post-intervention. There was a significant main effect for day ($F$ [1,11] = 12.1; $P$ = 0.005) in TS as well as a main effect over time ($F$ [9,99] = 40.3; $P$<0.001). There was a significant main effect for day ($F$ [1,11] = 7.0; $P$ = 0.02) as well as a main effect across time ($F$ [9,99] = 63.0; $P$<0.001) indicated in TC.

*Exercise performance.* Post 90 min incremental time trial performance increased by 44% (5.38±5.54 to 7.74±3.92min$^{-1}$; $P$ = 0.04; $d$ = 2.31: Very Large) pre- to post-intervention. PPO output increased by 55% (137±128 to 213±77W; $P$ = 0.03; $d$ = 3.39: Very Large). Fig 7

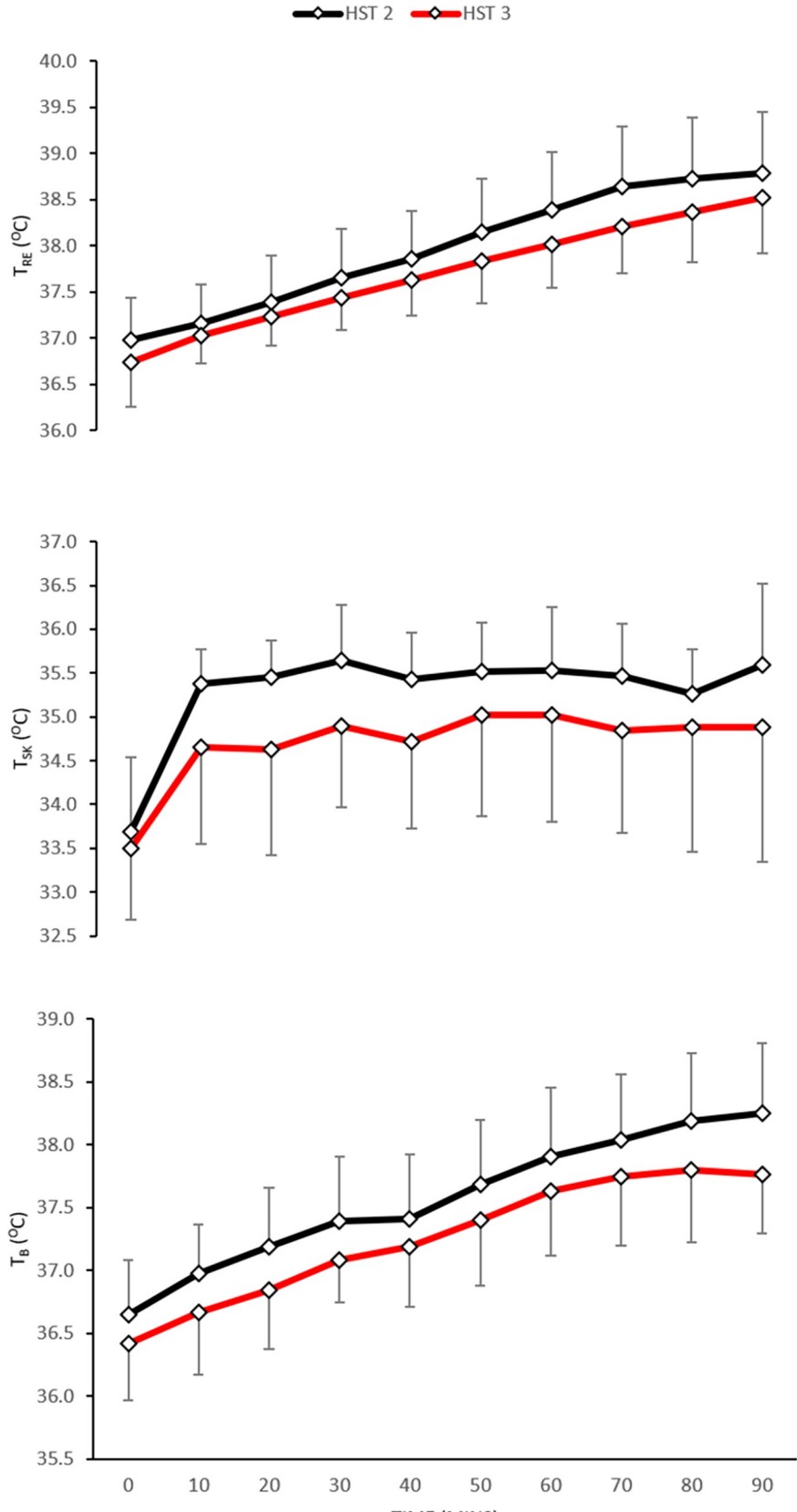

**Fig 5. Mean (SD) for HR pre- to post- short-term heat acclimation in hot conditions (35˚C, 60%RH).**

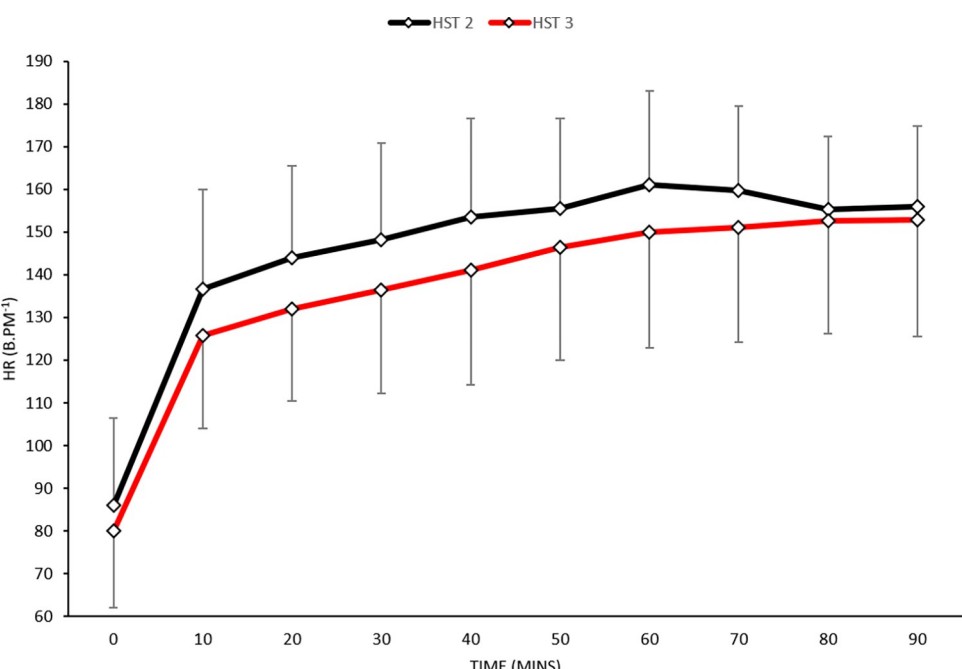

**Fig 6.** Mean (SD) for perceptual measures RPE (upper), TS (middle), and TC (bottom) pre- to post- short-term heat acclimation in hot conditions (35˚C, 60%RH).

represents the percentage completion of the HST and ramped protocol to exhaustion. Mean 90 min steady-state exercise performance increased by 8% (83.33±10.73 to 90.00±0.00min$^{-1}$; $P = 0.05$; $d = 4.12$: Very Large).

## Discussion

This study investigated the effectiveness of four consecutive days of 90 min isothermic exercise-heat stress protocol on 12 moderately trained males. Post intervention RPE, TC, and TS significantly decreased at specific time points while TTE and PPO both significantly increased during the ramped test to exhaustion. Descriptive statistics showed mean resting $T_{re}$ and HR was lowered, however this was insignificant. These findings suggest this protocol could be an effective tool for athletes preparing for exercise in the heat and may provide an effective supplement to warm weather training camps.

### Effectiveness of short-term isothermic heat acclimation

**Relationship between adaptive response and exposure duration.**   The wide variety of protocols questions the duration and type of heat exposure imposed. The more obvious variation between studies is the number of daily exposures outlined by [33]. For example 4 d [38–40], 5 d [28, 36, 49], 6 d [50–52] and 7 d [16, 53] have been used previously. However, the less obvious variation between studies is the duration of the daily exposure. Garrett, Goosens [36] and Garrett, Creasy [28] employed a 90 min duration whereas other studies range from 30 min to 120 min [33], with one study imposing a 240 min duration over 5 days [52]. Therefore, when designing and implementing an acclimation protocol, is the number of daily exposures more important than the duration of the exposure itself or vice-versa.

Pryor, Minson [34] suggest exercise intensities and rest periods should be sport specific. Subsequently, duration and number of daily exposures should consider the type of sport too.

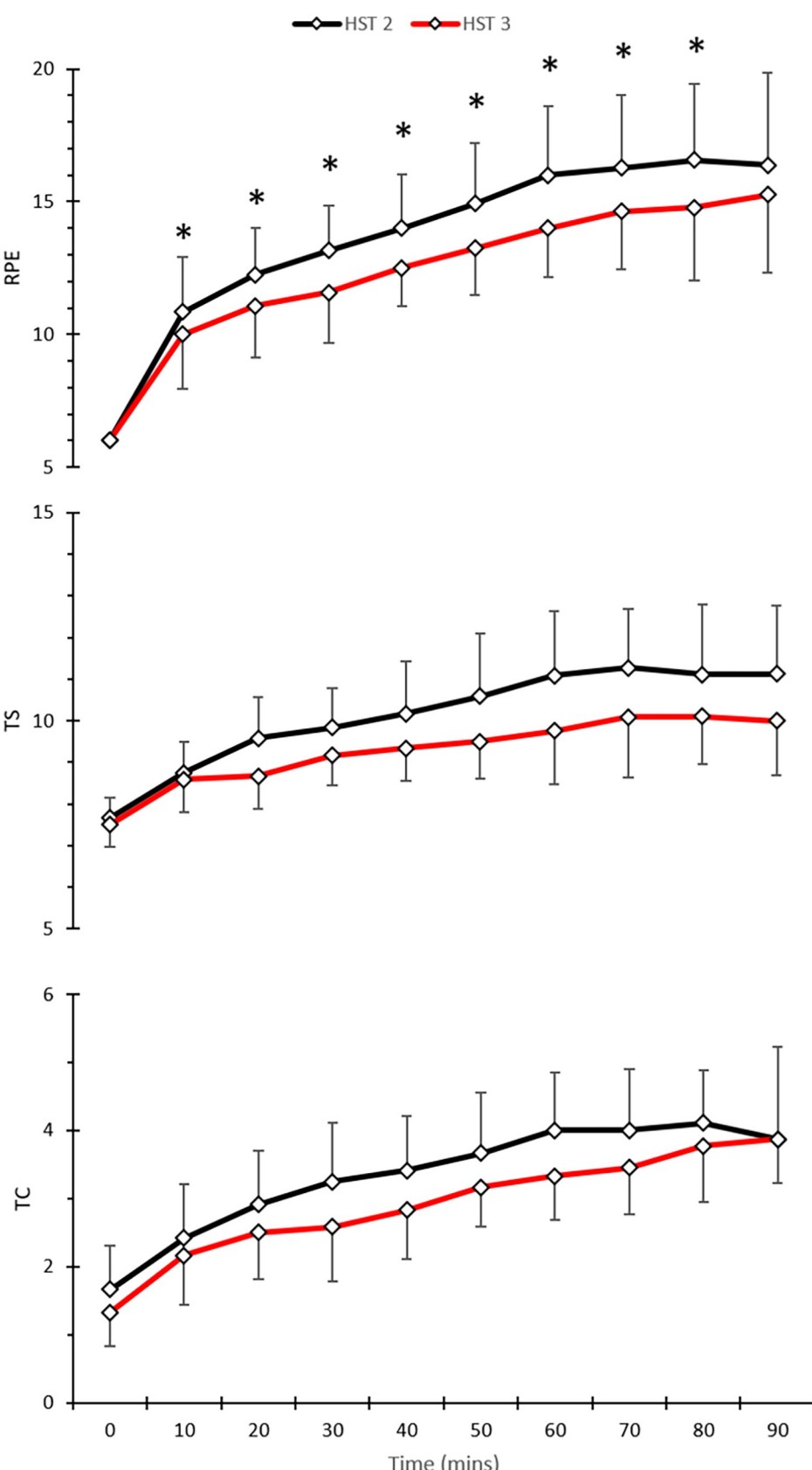

**Fig 7. Completion (%) of the 90 min (s) and ramped protocol to exhaustion (s) pre- to post- acclimation in hot conditions (35˚C, 60%RH).**

For example, if the participants are highly trained endurance athletes then it would be futile to design a short duration (<60 min) protocol with fewer daily exposures (<5 d)–due to training status as higher trained athletes already possess physiological and behavioural adaptations to cope with exercise-heat stress [2, 10, 28, 54]. Hence, it may be difficult to induce an adaptive stimulus to encourage adaptation [2, 34]. It may be practical to increase the duration >60 min and additional daily exposures to allow more time to induce a sufficient adaptive stimulus [2, 33, 34]. It is suggested to reduce the duration of heat exposure first, as the magnitude and rate of adaptations are greater in lesser trained individuals [2, 10, 28, 33, 34]. Therefore, shorter sessions of ≤60 mins over ≤5 d may be insufficient to induce physiological and perceptual benefits. In the current study, 90 min per day over 4 d was sufficient to encourage partial adaptation in moderately trained athletes. This was comparable with previous work over 5 d [2, 10, 28, 33].

Considering the duration and number of daily exposures, it may be worth noting the total exposure time versus the time spent above 38.5˚C $T_{re}$. A $T_{re}$ of 38.5˚C has been routinely used in controlled hyperthermia studies and is widely viewed as the threshold for adaptive response [2, 35, 49, 54, 55]. Therefore, it would seem appropriate to record the time spent above 38.5˚C when comparing between studies. In the present study, participants spent approximately 55% (3.3 hours) of total exposure time (6 hours) above 38.5˚C $T_{re}$ (Fig 2). A recent study by Moss, Bayne [2] only reports exposure time, where participants were heat exposed for a total of 5 h over 5 d. Similarly, previous work by Garrett, Creasy [28], Garrett, Goosens [10], Garrett, Gleadall-Siddall [49], and Neal, Corbett [54] implemented a total of 7.5 h over 5 d. Therefore, variations within study designs can make it difficult to compare [34], particularly with controlled hyperthermia methods, as the time spent above 38.5˚C $T_{re}$ may provide more information when comparing protocols and how different protocols affect the magnitude of heat adaptation.

**Rectal temperature.** The present protocol did not significantly reduce mean resting $T_{re}$ (-0.24; -0.54 to 0.07˚C). A lower resting $T_{re}$ is suggested to be an indicator of successful HA regimens, as it delays critical exercise-capacity limiting, high-core body temperatures in the heat [2, 3, 56]. Inducing a lower resting $T_{re}$ through HA is important in uncompensable heat stress by widening the core temperature band at the time of day heat-exercise bouts occur [57].

The present findings could be explained by greater reductions in resting $T_{re}$ by the lower training status of several participants compared to higher trained participants in this study–with previous literature suggesting minor reductions in resting $T_{re}$ occur in higher trained individuals [2, 10, 28, 54]. More efficient heat loss mechanisms at a lower core temperature add to the reduction in thermal strain [2], evidenced by the number of participants who did not complete the 90 min continuous exercise bout in HST2 but did in HST3 (Fig 7). This is similar to the findings of Mackay, Patterson [58] who found that 5 d of HA was adequate to improve time-to-exhaustion in a rugby league specific HA regimen–in temperate conditions. Compared to a previous 5 d study, greater changes were observed across a shorter period of time (45 min), as well as main effects between trials [2].

**Heart rate.** Mean resting HR was not significantly reduced by HA (-6; -19 to 6 b.min⁻¹). A recent meta-analysis inferred that HA offers moderate benefits on lowering resting HR [33], while current data indicates a negligible effect on resting HR (-6; -19 to 6 b·min⁻¹; $d = 0.32$: small). It is widely accepted that HR adaptations occur rapidly, typically within 4–5 d [33]. As

with $T_{re}$, participants' training status affects the magnitude of adaptation that occurs, with lesser trained individuals undergoing a greater degree of adaptation [10, 33].

Exercising HR was stabilised across the 90 min. A similar 5 d STHA protocol by Moss, Bayne [2] demonstrates a significant difference at similar time-points at 20, 30, and 40 min having only recorded 45 min of exercise. It is suggested that HA does not improve the retention of PV during exercise in the heat [10, 54, 57, 59]. The current study found PV decreased further by 41% (-4.70±10.88 to -6.63±10.71%; $P = 0.62$; $d = 0.84$: Moderate). This is despite a 16% increase from day one to four of acclimation (-5.78±6.55 to -4.85±11.39%; $P = 0.82$; $d = 0.38$: Small). Typically, increased HR stability is accompanied by PV expansion which aids with the reduction of CV strain during exercise-heat bouts [2, 33]. Data from the meta-analysis indicates that PV expansion is common following HA regardless of methodology [33]. However, this is not the case in the current study where mean PV decreased contradicting earlier work. Training status of participants has been known to affect the magnitude of adaptation, as higher trained, endurance athletes already possess expanded PV [57]. This may be attributed to the type of sport athletes partake in which may impose varying physical demands.

Schleh, Ruby [60] describe the relationship between PV expansion and the hormone [aldo]$_p$ and they observed a significant decrease in [aldo]$_p$ levels both pre- and post-exercise following acclimation contradicting that HA would promote the release of the hormone enabling PV expansion. They attributed this to an already increased PV at rest and therefore limited further expansion [60]. The current study contradicts the work of Schleh, Ruby [60] as increases in resting and post-exercise [aldo]$_p$ concentrations on day four were significantly increased. Similar work conducted by Garrett, Goosens [10] found a comparable relationship between increases in aldosterone and PV expansion, suggesting a positive feedback mechanism because of increasing plasma osmolality caused by dehydration on [aldo]$_p$ [10, 60].

**Perceptual adaptations.** Participants felt more thermally comfortable and perceived reduced strain post-HA consistent with previous work [2]. The current perceptual data indicates the 4 d programme can be just as effective as previous 5 d, although longer durations of >7 d offer an increased magnitude of adaptations [2].

Reducing the perception of effort can potentially increase the capacity of exercise, particularly in subsequent heat-exercise bouts. It is suggested that a lower RPE enables participants to tolerate prolonged steady-state exercise [33]. Data showing the lowering of RPE was only present in three studies [33]. However with the addition of Moss, Bayne [2], and the current study, RPE is shown to be lowered consistently throughout STHA (4–7 d). Willmott, Hayes [40] observed a lowering in RPE with concurrent reductions in $T_{re}$ and partly attribute performance improvements to improved comfort levels related to lower RPE and the same fixed exercise intensity in the respective groups' final sessions. The perceptual adaptations found in previous STHA work could be representative of a reduced tendency to select lower exercise intensities in the heat, as well as possibly maintaining decision-making and cognitive functions during race conditions [40, 61].

**Performance and time to exhaustion trial.** Training status of the participants may vary within the group. Those who are lesser trained may have experienced a greater magnitude of physiological adaptation compared to their higher trained counterparts [2, 10, 28, 54]. Additionally, the lesser trained participants may have adapted their behaviour more radically as they may have been less familiar and experienced with the demand of endurance exercise. Whereas those who are endurance trained possess some adaptations before they undergo acclimation [2, 10, 28, 33, 34]. Both statements reflect previous work that physiological adaptations provide an increased buffer before reaching performance compromising core temperatures [57] and perceptual/behavioural adaptations enhance participant performance during steady-state exercise performance in the heat [33]. Continuous exercise performance demonstrated

improvement post-acclimation in the hot conditions of the HSTs (35˚C; 60%RH) and improved to the extent that all participants completed the 90 min steady state exercise post intervention compared to the 7 participants pre-intervention.

The incremental protocol to exhaustion reported significant increases in TTE ($P = 0.04$) and PPO ($P = 0.03$). Due to the varying methodologies used in the application of HA [34], it is difficult to compare directly to other performance data. However, other previous STHA ($\leq 7$ d) performance data indicate improved performance capability in varying hot conditions. Garrett, Creasy [28] observed a mean decrease of 4 s in time to completion in untrained and moderately trained males ($P = 0.02$). Meanwhile, Garrett, Goosens [36] observed a 106 s increase in TTE post-STHA (5 d) ($P = 0.001$) using an identical protocol to this study but with 5 d. Conversely, Neal, Corbett [54] found no increase in time to completion ($P = 0.38$) but did observe an increase in mean PPO ($P = 0.056$) albeit this was deemed insignificant.

## Fluid retention

The fundamental component of using the controlled hyperthermia technique is that participants experience the same thermal load, as was the case in the current study—experiencing mild hypohydration of ~1.3% BM in both the pre- and post-intervention trials (Table 1). Blood parameters during acclimation (Table 3) indicates a meaningful change in $[Na^+]_p$–which is responsible for aiding fluid retention [35]–on day one but not day four despite an increase in resting $[aldo]_p$. Post-exercise $[aldo]_p$ increased significantly from resting concentrations on day four. An increase in $[aldo]_p$ concentrations without increases in $[Na^+]_p$ is inconsistent with previous data [35, 62–64] but is not universal [28]. This is unexpected as the notable effects of $[aldo]_p$ is the retention of $[Na^+]_p$, thereby retaining water from the urine to maintain extracellular volume and, consequently, blood volume [28, 35]. [28] observed a PV increase without concurrent $[Na^+]_p$ increases which was attributed to the increase in PV itself. As PV had already increased (4.5%), it may not have been a necessary function to attempt to increase further. Interestingly, the absence of such a change on day four similar to day one when considering $[aldo]_p$ and PV measures could be related to the reduced heat exposure compared to previous studies [35, 62–64].

## Cortisol

Cortisol is frequently described as an indicator of physical and psychological strain [2]. While this study did not find any significant differences in cortisol, a greater mean within-trial increase occurred pre- to post-exposure on day four compared to day one with post-exercise measures being very similar (Table 3). This observation conflicts with Watkins, Cheek [24] and Moss, Bayne [2] who found significant within trial increases, specifically in pre-intervention measures where baseline cortisol was not different–and participants were allowed to drink *ad libitum* either before and after [24] or during [2] heat exposure. Costello, Rendell [65] also observed a similar increase in pre-intervention cortisol levels, however this was in a dehydrated state and was not replicated when hydration was maintained.

Costello, Rendell [65] offer three plausible explanations for their findings. The first being circadian rhythm and cortisol decreasing naturally as the biological day progresses. The second being a reduced catabolic response to the same pre-training exercise stimulus or stress, likely augmented by HA. The third being increases in circulating cortisol levels being visible after exercise if the intensity is $\geq 60\%$ of maximal oxygen consumption for 20 min or more. In the context of this study, trials were completed at the same time of day for every visit. Catabolic responses by the definition offered may not apply as the current post intervention within-trial increase was greater than pre-intervention. Finally, the experimental design was 90 min and

cortisol measures obtained during the intervention period as opposed to pre-post intervention HSTs.

Based on the present data, we speculate that cortisol behaved similarly to $T_{re}$, widening the range between baseline and end-exercise. We observed a concurrent reduction in mean subjective measures in pre-post HSTs (Fig 6) and increased mean exercise duration (Fig 7), and with cortisol being frequently described as measure of physiological and psychological stress, lower baseline cortisol may be the cause with participants being less stressed prior to exercising in the heat.

### Heat shock proteins

The present study found limited changes in HSP70/TP response to exercise-heat stress consistent with previous work–albeit data is very limited–which suggests HA has a trivial effect on increasing extracellular concentrations of HSP70 [33]. However, increases of 110% [66] and 320% [67] have been reported [33]. The present data suggests HA has very little effect on HSP concentrations as an adaptive response in shorter acclimation protocols.

### Limitations

Overall, 12 participants took part in this study with n = 11 completing the control trial and n = 12 completing the intervention trial. We controlled for heat but not exercise. Testing periods were conducted outside of the British Summer Time and participants were instructed to follow their typical training regime whilst refraining from exercising 24h prior to any visit. Current literature suggests this effect is minimal but could be more influential on lesser trained individuals. Data and implications cannot be generalised beyond males as participants were from various sporting backgrounds, ages, and training status. A more homogenous group would be more beneficial in future. Sample sizes in this field are an issue in general, with groups of 3–4 participants taking 4–5 weeks depending on methodology. Our sample size ($n$ = 12) is larger-than-most compared to previous work. Participants were sourced from volunteers based at or associated with the University of Hull, recruited on a convenience basis. Participants are also from a variety of sporting backgrounds (cycling, football, rugby, and running) and performance levels (PL1: $n$ = 5; PL2: $n$ = 2; PL3: $n$ = 5). Future studies would benefit from a more homogenous group of participants from similar sports and training status.

### Conclusion

In summary, short-term isothermic HA (4 d) with no fluid intake enhanced performance capacity in hot and humid conditions in moderately trained males. Restricting fluid intake was adequate in increasing the physiological and perceptual stress experienced by participants thereby enhancing adaptation to the environment. Regardless of the level of physiological adaptations, the behavioural adaptations coupled with the perceptual benefit brought about by 4 d isothermic STHA was enough to elicit improved performance and thermotolerance in hot conditions. In terms of cellular heat stress response, additional exposures may be required to ensure this level of adaptation.

### Author Contributions

**Data curation:** Andrew J. Simpson, Rebecca V. Vince.

**Investigation:** Jake Shaw, Cory Walkington, Edward Cole.

**Methodology:** Jake Shaw, Andrew T. Garrett.

**Resources:** Damien O. Gleadall-Siddall, Rachel Burke, James Bray, Andrew T. Garrett.

**Supervision:** James Bray, Andrew J. Simpson, Andrew T. Garrett.

**Writing – original draft:** Jake Shaw.

**Writing – review & editing:** Jake Shaw, Damien O. Gleadall-Siddall, Rachel Burke, Andrew J. Simpson, Andrew T. Garrett.

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
