## [Decision Letter · Decision Letter 0]

19 Oct 2021

PONE-D-21-25075Effectiveness of Short-Term Isothermic-Heat Acclimation (4 days) on Moderately Trained MalesPLOS ONE

Dear Dr. Jake Shaw,

Thank you for submitting your manuscript to PLOS ONE. After careful consideration, we feel that it has merit but does not fully meet PLOS ONE’s publication criteria as it currently stands. Therefore, we invite you to submit a revised version of the manuscript that addresses the points raised during the review process.

Two experts in the field evaluated the present study. Although the manuscript provides exciting data and may potentially be suitable for publication in PLOS One, it still needs substantial revision and improvement. Remarkably, as stated by the second reviewer, the methods section is incomplete, thus hampering the understanding of the study design and current findings. Please take every reviewer's comment very seriously, and please also address the editor's comments at the end of this letter.

We look forward to receiving your revised manuscript.

Kind regards,

Samuel Penna Wanner, Ph.D.

Academic Editor

PLOS ONE

Journal Requirements:

2. Please note that in order to use the direct billing option the corresponding author must be affiliated with the chosen institute. Please either amend your manuscript to change the affiliation or corresponding author, or email us at plosone@plos.org with a request to remove this option.

Additional Editor Comments:

The authors must significantly improve their manuscript. It was challenging to understand the methods and, therefore, to follow the results. The manuscript has nine authors and will benefit from a critical reading by all of them. Please see some comments below; these and other issues should be addressed to make the manuscript suitable for publication.

1- The short title has to be shorter than the full title.

2- The full title should indicate that the authors have investigated the effectiveness of short-term heat acclimation on physical performance.

3- Abstract: There is no information on how physical performance was evaluated. Moreover, several abbreviations were not defined before their first appearance. In general lines, I would suggest focusing the abstract on the study's main findings and avoid describing all the parameters measured.

4- Methods. Please explain why the participants were characterized as moderately trained subjects. Please include information about their training (i.e., cycling) experience. Maybe, it would be interesting to classify the subjects according to the study of De Pauw et al. Int J Sports Physiol Perform, v.8 (n.2): p.111-122, 2013 (doi: 10.1123/ijspp.8.2.111).

5- The methods section is incomplete. Please see some examples as follows: How did the authors measure the plasma concentrations of aldosterone, cortisol, albumin, and heat shock protein 72? What is the rationale for performing these measurements? Please indicate when blood samples were collected? Were these plasma concentrations corrected for exercise-induced changes in plasma volume?

6- There are sentences in which the authors mention the heat shock protein 72; in contrast, they mention HSP70 in Table 3. Please standardize this throughout the manuscript. Moreover, how was total protein measured?

7- Unless I have overseen this information, the methods section does not describe the measurement of perceptive parameters and the plasma concentration of sodium.

8- Did the authors calculate the ICC for the measured parameters, particularly for the performance parameters? This information can be obtained using the HST1 and HST2 and is quite relevant considering the marked improvement in physical performance observed after only four days of heat acclimation? It would be essential to clarify that heat acclimation-induced changes in performance are real and not the result of measurement error.

9- The figure showing the experimental timeline can be improved in several ways. Please see some examples as follows: a- to indicate that an incremental test followed the heat stress test; b- to indicate the parameters measured during the heat stress tests; c- what does dehydration mean in the context of this study?

10- The authors may want to use colored symbols in their figures to identify the data from HST2 and HST3. Please note that PLOS One manuscripts are only published online.

11- The data in figure 6 can be analyzed using the log-rank test (Bland and Altman. BMJ, v.328 (n.7447): p.1073, 2004; doi: 10.1136/bmj.328.7447.1073). This analysis has been used previously to evaluate physical performance in tests to fatigue/exhaustion (Wanner et al. Int J Biometeorol, v.58: p. 1077-1085, 2014; doi: 10.1007/s00484-013-0699-y).

12- Table 2 and 3. What do the p-values mean in these tables? Do they mean the main effect of exercise sessions or heat acclimation? Because the authors have performed two-way ANOVAs, they can present three p-values in these tables: exercise session, heat acclimation, and exercise session x heat acclimation interaction.

13- Table 2. There is exaggerated precision for urine-specific gravity (i.e., four decimal places). Please considering using fewer decimal places for this parameter.

This academic editor is looking forward to receiving a revised and improved version of the manuscript.

Reviewers' comments:

Reviewer's Responses to Questions

**Comments to the Author**

1. Is the manuscript technically sound, and do the data support the conclusions?

Reviewer #1: Yes

Reviewer #2: Partly

2. Has the statistical analysis been performed appropriately and rigorously? 

Reviewer #1: Yes

Reviewer #2: No

3. Have the authors made all data underlying the findings in their manuscript fully available?

Reviewer #1: Yes

Reviewer #2: Yes

4. Is the manuscript presented in an intelligible fashion and written in standard English?

Reviewer #1: Yes

Reviewer #2: Yes

5. Review Comments to the Author

Reviewer #1: Dear authors, congratulations on your study, which was carefully conducted and present adequate controls. The following comments are aimed to clarify and improve the manuscript, including information that could be added in the revised version.

How did you test 12 individuals? Did the authors made a priori sample size calculation (if so, please add this calculation)? Alternatively, was this sample size obtained as a convenience, or was it based on the number of participants in similar studies?

Were the experimental procedures carried out using each participant's own bicycle or did the researchers provide one same bicycle for all of them?

Since generally trained people exercise outdoors, did you familiarized the subjects with exercising indoors? If so, please inform the duration, speed/intensity and environmental conditions during the familiarization sessions. If not, please justify why this familiarzation was not required.

Because many people are not familiar with the perception scales, were the subjects previously familiarized with their use?

Please include a paragraph with the limitations of your study.

The results indicate that a short-term acclimatization period induced several adaptations, either physiological or perceptual adaptations. However, this reviewer suggests that future experiments should include a performance-oriented test (in addition to heat-stress tests), such as the countermovement jump. For example, the countermovement jump which could be used could be performed before and after heat stress testing session to evaluate whether there was a decrease in lower limb power induced by the test and whether acclimation will influence this decrease.

Reviewer #2: The authors completed a 4 day, 90 min isothermic HA study, with heat stress tests completed before, and following, the HA period. I have found the study design to be vague and challenging to understand and have considerable reservations regarding some of the statistical methods. In short, I believe that the authors should consider expanding the methodological section, rechecking the statistical outcomes and considerably tempering language used – the outcomes of this study are weakened by the lack of a thermoneutral/temperate control group, yet this is not discussed. There are also quite a few issues with spelling and grammar throughout the paper.

Please see reviewer comments attachment doc for more specific information

6. PLOS authors have the option to publish the peer review history of their article (what does this mean?). If published, this will include your full peer review and any attached files.

Reviewer #1: **Yes: **Jefferson Fernando Coelho Rodrigues Junior

Reviewer #2: **Yes: **John Owen Osborne

---

## [Author Response · Author response to Decision Letter 0]

24 Feb 2022

Please see the attached JSHAWPLOSONERESPONSE document for detailed responses to each and every comment provided.

---

## [Decision Letter · Decision Letter 1]

10 Mar 2022

PONE-D-21-25075R1Effectiveness of Short-Term Isothermic-Heat Acclimation (4 days) on Physical Performance in Moderately Trained MalesPLOS ONE

Dear Dr. Shaw,

Thank you for submitting your manuscript to PLOS ONE. After careful consideration, we feel that it has merit but does not fully meet PLOS ONE’s publication criteria as it currently stands. Therefore, we invite you to submit a revised version of the manuscript that addresses the points raised during the review process. As the authors will see in the comments below, both reviewers were supportive of the changes that have been made to the manuscript. However, whereas Reviewer #1 had not further recommendations, Reviewer #2 highlighted some important points that still require changes and/or clarifications. After reading the feedback from Reviewer #2, I feel that the authors should be able to satisfactorily address the comments. However, please note that the revisions are still qualified as major at this stage in the process.

We look forward to receiving your revised manuscript.

Kind regards,

Juan M. Murias

Academic Editor

PLOS ONE

Reviewers' comments:

Reviewer's Responses to Questions

**Comments to the Author**

1. If the authors have adequately addressed your comments raised in a previous round of review and you feel that this manuscript is now acceptable for publication, you may indicate that here to bypass the “Comments to the Author” section, enter your conflict of interest statement in the “Confidential to Editor” section, and submit your "Accept" recommendation.

Reviewer #1: All comments have been addressed

Reviewer #2: (No Response)

2. Is the manuscript technically sound, and do the data support the conclusions?

Reviewer #1: Yes

Reviewer #2: Partly

3. Has the statistical analysis been performed appropriately and rigorously? 

Reviewer #1: Yes

Reviewer #2: No

4. Have the authors made all data underlying the findings in their manuscript fully available?

Reviewer #1: Yes

Reviewer #2: Yes

5. Is the manuscript presented in an intelligible fashion and written in standard English?

Reviewer #1: Yes

Reviewer #2: Yes

6. Review Comments to the Author

Reviewer #1: Dear authors, congratulations on your study, which was carefully conducted and present adequate controls. The study presented robust methodological data well designed with a design. The questions were answered, in view of that -- -------- -- -- ------ --- ----------.

Reviewer #2: General Comments

I commend the authors on the considerable changes and resultant improvement in the manuscript. However I still have some questions around the results.

Abstract and Introduction

Line 43: why is an exact P value provide in line 42, but P<0.05 used here? Same with the lines further on in the abstract.

I really enjoyed reading the revised/improved introduction. Good work.

Methods

Good job improving the explanation of the study design. I believe it is much clearer and easier to understand now.

Line 161: I believe the word should be ‘inverted’ not ‘invested’?

I previously suggested that the use of ‘cardiac frequency’ is rare and potentially confusing to readers. This was agreed in the response to my comments, but this change has not been undertaken throughout the document?

Line 173: Apologises if I wasn’t clear in my previous comment, but I meant to please calculate and provide the intraassay CV (and inter- if you used multiple plates for the same variable) of each bloods variables from the ELISA plates.

Line 199: I am pretty sure this should say ‘6-20’ scale, not 1-20?

Data analysis section: how was missing data handled?

Results

Based on what is written in the HST section of the results, Tre and HR saw no main effects for day or an interaction, just time. This makes sense. But then the subsequent sentences (Line 248-250 and 257-259) discuss that rest and end-exercise Tre and HR were reduced…and seems this means ‘reduced compared to Day 1’. Which is not what is described in the analysis, as it has already been indicated that there was no interaction effect? So really it should only be demonstrating differences in time, e.g., rest vs endpoint (which will obviously be sig. different). So I am a little confused on what these comparison are referring to?

This is highly important as it is a central premise of this paper. For example Line 319-20 and 338-340 in the discussion.

Line 247 – states Fig 5 upper, think it should be Fig 4

Discussion and Conclusion

As stated above, I do not fully understand about the statements at the start of the Tre and HR sections, regarding resting measures. For example, the reductions in resting Tre mentioned in Line 324 appear to refer to significant reduction in this measure between pre- and post-HA (i.e., effect of the HA intervention). But in the results section this is discussed as if it was a t-test of rest and 90min pre-HA vs post-HA, yet the preceding sentences specify that there was only a main effect for time (and not day x time).

The large variance of both of these resting changes do not exactly support the theory that there was a true and meaningful difference? E.g., Tre resting change was -0.24C but had a variance of double this (0.48C). This is also observable in the HR section.

Maybe I am misunderstanding some of the analysis that was undertaken?

The cortisol section is very interesting.

The limitation section is insufficient. I strongly encourage the authors to consider all potential limitations with this present study. For example, the heterogeneity of the type of athletes (endurance, teamsport, strength etc) and indeed the considerable range in performance level (n=5 were DP1 which is ‘untrained’, while n = 5 were ‘trained’!). Acknowledge of these issues does not devalue the importance of the research, rather it allows a reader to gain a better insight into the study outcomes and comparison to other literature.

7. PLOS authors have the option to publish the peer review history of their article (what does this mean?). If published, this will include your full peer review and any attached files.

Reviewer #1: No

Reviewer #2: **Yes: **John O. Osborne

---

## [Author Response · Author response to Decision Letter 1]

23 Apr 2022

Reviewer 1: We thank you for your critique on our manuscript and the words of encouragement in the last round of revisions.

Reviewer 2: We thank you for your thorughness with our manuscript.

---

## [Decision Letter · Decision Letter 2]

10 May 2022

PONE-D-21-25075R2Effectiveness of Short-Term Isothermic-Heat Acclimation (4 days) on Physical Performance in Moderately Trained MalesPLOS ONE

Dear Dr. Shaw,

Thank you for submitting your manuscript to PLOS ONE. After careful consideration, we feel that it has merit but does not fully meet PLOS ONE’s publication criteria as it currently stands. Therefore, we invite you to submit a revised version of the manuscript that addresses the points raised during the review process. Although one reviewer is satisfied with the responses, the other reviewer still has some requests that need to be properly addressed. Specifically, the reviewer highlighted the presence of half finished sentences, misspelled words, incorrect references to figures, old data that were not removed, or paragraphs that discussed results that were not provided. This makes the manuscript difficult to read and the reviewer's task very challenging. The authors should note that it is not the reviewer's role to fix this type of issues. The reviewer has provided some specific examples of components of the manuscript that should be changed. I would encourage the authors to address these comments appropriately with the goal of finishing this review process effectively. To do so, the author should pay extreme attention to details in the updated version. Similarly, the authors might want to consider looking for external help with the editing process if they are not able to fix the issues that the reviewer highlighted. Otherwise, I will need to request that a third reviewer is involved, which will result in substantial delays in the process.

We look forward to receiving your revised manuscript.

Kind regards,

Juan M. Murias

Academic Editor

PLOS ONE

Reviewers' comments:

Reviewer's Responses to Questions

**Comments to the Author**

1. If the authors have adequately addressed your comments raised in a previous round of review and you feel that this manuscript is now acceptable for publication, you may indicate that here to bypass the “Comments to the Author” section, enter your conflict of interest statement in the “Confidential to Editor” section, and submit your "Accept" recommendation.

Reviewer #1: All comments have been addressed

Reviewer #2: (No Response)

2. Is the manuscript technically sound, and do the data support the conclusions?

Reviewer #1: Yes

Reviewer #2: Partly

3. Has the statistical analysis been performed appropriately and rigorously? 

Reviewer #1: Yes

Reviewer #2: Yes

4. Have the authors made all data underlying the findings in their manuscript fully available?

Reviewer #1: Yes

Reviewer #2: Yes

5. Is the manuscript presented in an intelligible fashion and written in standard English?

Reviewer #1: Yes

Reviewer #2: No

6. Review Comments to the Author

Reviewer #1: The article contemplated all the considerations raised during the review. The article meets all requirements, without further suggestions.

Reviewer #2: Dear authors,

Thank you for the detailed responses and comments. I appreciate that my feedback has been useful.

My main issue remains: there is language used within the manuscript, which I do not believe that the results support. For example, Line 289-292. 'post intervention Tre, fc (should be HR), RPE, TC and TS significantly decreased at specific timepoints...' yet based on the results in Lines 252-258, there were only significant effects for time (not day or interaction). Similar data is presented for HR. This is further evidenced in the Figures 4 and 6 which should no pairwise differences. So I question where the sig. differences of Line 289-290 are?

I raised this issue in a previous comment (response #7 and #9) but the author response did not assuage my concerns. I appreciate that the descriptive data were kept, but it should not be discussed as it there were sig.differences between these physiological values across the intervention.

Line 257 - remove the word 'in' after the bracket.

Line 122 and 289 - fc should be HR

Line 303 - remove '?'. Also this sentence doesn't read particularly well with the preceding sentence.

Line 336 - refers to Fig 3 but this appears to be the wrong figure?

The abstract mentions results and findings that aren't discussed in the manuscript? E.g., at 70mins, Tre was low pre- to post intervention?? Similarly, the conclusion to the abstract is incorrect based on the present manuscript.

The limitations paragraph is somewhat challenging to understand in the present format; it seems a little disjointed. For example 'we controlled for heat but not exercise' - I assume this refers to the lack of a training control maybe? Needs to be expanded upon. The next sentence also needs clarification. The final sentence doesn't even finish correctly, it just ends partway through a name: University of....? I would suggest that this paragraph be revised to enhance the readability.

I hope these comments have been of some help.

7. PLOS authors have the option to publish the peer review history of their article (what does this mean?). If published, this will include your full peer review and any attached files.

Reviewer #1: **Yes: **Jefferson Fernando Coelho Rodrigues Junior

Reviewer #2: **Yes: **John Owen Osborne

---

## [Author Response · Author response to Decision Letter 2]

13 May 2022

Reviewer #2: We greatly appreciate your patience. We have taken your point on board regarding the language and hope the changes we have made are to your satisfaction. Please see our document entitled "JSHAWPLOSONERESPONSE" for detailed responses to all of your comments.

---

## [Decision Letter · Decision Letter 3]

6 Jun 2022

Effectiveness of Short-Term Isothermic-Heat Acclimation (4 days) on Physical Performance in Moderately Trained Males

PONE-D-21-25075R3

Dear Dr. Shaw,

We’re pleased to inform you that your manuscript has been judged scientifically suitable for publication and will be formally accepted for publication once it meets all outstanding technical requirements.

Kind regards,

Juan M. Murias

Academic Editor

PLOS ONE

Additional Editor Comments (optional):

Reviewers' comments:

Reviewer's Responses to Questions

**Comments to the Author**

1. If the authors have adequately addressed your comments raised in a previous round of review and you feel that this manuscript is now acceptable for publication, you may indicate that here to bypass the “Comments to the Author” section, enter your conflict of interest statement in the “Confidential to Editor” section, and submit your "Accept" recommendation.

Reviewer #2: All comments have been addressed

2. Is the manuscript technically sound, and do the data support the conclusions?

Reviewer #2: Yes

3. Has the statistical analysis been performed appropriately and rigorously? 

Reviewer #2: Yes

4. Have the authors made all data underlying the findings in their manuscript fully available?

Reviewer #2: Yes

5. Is the manuscript presented in an intelligible fashion and written in standard English?

Reviewer #2: Yes

6. Review Comments to the Author

Reviewer #2: (No Response)

7. PLOS authors have the option to publish the peer review history of their article (what does this mean?). If published, this will include your full peer review and any attached files.

Reviewer #2: **Yes: **John Owen Osborne

---

## [Editor Report · Acceptance letter]

8 Jun 2022

PONE-D-21-25075R3 

Effectiveness of Short-Term Isothermic-Heat Acclimation (4 days) on Physical Performance in Moderately Trained Males 

Dear Dr. Shaw:

I'm pleased to inform you that your manuscript has been deemed suitable for publication in PLOS ONE. Congratulations! Your manuscript is now with our production department. 

Kind regards, 

on behalf of

Dr. Juan M. Murias 

Academic Editor

PLOS ONE